# Structural and mechanistic insights into regulation of the retromer coat by TBC1d5

Da Jia[1,2], Jin-San Zhang[3,4], Fang Li[1], Jing Wang[1], Zhihui Deng[3,5], Mark A. White[6], Douglas G. Osborne[3], Christine Phillips-Krawczak[3], Timothy S. Gomez[3], Haiying Li[7], Amika Singla[7], Ezra Burstein[7,8], Daniel D. Billadeau[3] & Michael K. Rosen[2,9]

Retromer is a membrane coat complex that is recruited to endosomes by the small GTPase Rab7 and sorting nexin 3. The timing of this interaction and consequent endosomal dynamics are thought to be regulated by the guanine nucleotide cycle of Rab7. Here we demonstrate that TBC1d5, a GTPase-activating protein (GAP) for Rab7, is a high-affinity ligand of the retromer cargo selective complex VPS26/VPS29/VPS35. The crystal structure of the TBC1d5 GAP domain bound to VPS29 and complementary biochemical and cellular data show that a loop from TBC1d5 binds to a conserved hydrophobic pocket on VPS29 opposite the VPS29–VPS35 interface. Additional data suggest that a distinct loop of the GAP domain may contact VPS35. Loss of TBC1d5 causes defective retromer-dependent trafficking of receptors. Our findings illustrate how retromer recruits a GAP, which is likely to be involved in the timing of Rab7 inactivation leading to membrane uncoating, with important consequences for receptor trafficking.

[1] Key Laboratory of Birth Defects and Related Diseases of Women and Children (Ministry of Education), Department of Paediatrics, West China Second University Hospital and State Key Laboratory of Biotherapy, Sichuan University, Chengdu 610041, China. [2] Department of Biophysics, UT Southwestern Medical Center, 5323 Harry Hines Boulevard, Dallas, Texas 75390, USA. [3] Departments of Immunology, Biochemistry and Molecular Biology, Mayo Clinic College of Medicine, Mayo Clinic, Rochester, Minnesota 55905, USA. [4] Key Laboratory of Biotechnology and Pharmaceutical Engineering, School of Pharmaceutical Sciences, Wenzhou Medical University, Wenzhou, Zhejiang 325035, China. [5] Department of Pathophysiology, Qiqihar Medical University, Qiqihar 161006, China. [6] Sealy Center for Structural Biology, University of Texas Medical Branch, Galveston, Texas 77555, USA. [7] Department of Internal Medicine, UT Southwestern Medical Center, Dallas, Texas 75390, USA. [8] Department of Molecular Biology, UT Southwestern Medical Center, Dallas, Texas 75390, USA. [9] Howard Hughes Medical Institute, UT Southwestern Medical Center, Dallas, Texas 75390, USA. Correspondence and requests for materials should be addressed to D.J. (email: Jiada@scu.edu.cn) or to D.D.B. (email: Billadeau.Daniel@mayo.edu) or to M.K.R. (email: Michael.Rosen@utsouthwestern.edu).

Selective transport between membrane-bound organelles and between organelles and the plasma membrane is fundamental to cellular processes ranging from protein and lipid homeostasis to cell signalling[1,2]. Protein machineries known as coat protein complexes play central roles in selective transport by packaging specific membrane-bound cargoes into vesicles and tubules, and delivering them to specific organelles[1,2]. Much of our understanding of vesicle transport comes from studies of three classes of coats: Clathrin/Adaptor protein, COPI and COPII. A central concept that has emerged from this work is that small GTPases play important roles during multiple steps of vesicle formation, including both coat recruitment and vesicle maturation. Both Clathrin/Adaptor protein-1 and COPI are recruited to membranes by the Arf1 GTPase[3,4]. Similarly, the COPII coat is recruited to the endoplasmic reticulum membranes through the interaction between its subunit Sec23 and the Sar1 GTPase[5,6]. Both COPI and COPII also contain or bind GTPase-activating proteins (GAPs) that accelerate hydrolysis of GTP to GDP in their cognate GTPases, an event that triggers release of the coats from membranes: COPI binds to the ARF1 GAP[7,8], which triggers hydrolysis of GTP on ARF1 and the Sec23 subunit of COPII is a Sar1 GAP[5]. This mechanism, where a coat directly recruits a factor that promotes its dissociation from membranes, is believed to afford precise timing over the coating and uncoating processes during vesicle trafficking.

Retromer is a distinct class of coat protein, which bears no obvious sequence or structural similarity with the above three classes of well-studied coats[9–11]. Retromer is evolutionarily conserved across all eukaryotes and mediates cellular trafficking from endosomes to the *trans*-Golgi network (TGN) or to the plasma membrane[11,12]. Many endosomal proteins with a variety of functions depend on retromer for cellular transport, including sorting receptors (such as CI-M6PR), αvβ1 integrin, ion and copper transporters and SorL1, a receptor for amyloid precursor protein[9,11,12]. Retromer is required for proper lysosomal function, establishing Wnt gradients, maintaining metabolite homeostasis and many other cellular processes[9,11,13]. Defects in retromer have been associated with a wide range of human diseases, including Alzheimer's disease, Parkinson's disease, and bacterial and viral infection[9–11].

Retromer is composed of two subcomplexes, a cargo selection complex (CSC), consisting of subunits VPS35, VPS29 and VPS26, and an SNX (sorting nexin) module that binds to and remodels membranes. The SNX module can be either a dimer of SNX1/2 and SNX5/6, which contain BAR domains and associate with tubular membranes, or SNX3, which does not have a BAR domain. Similar to other classes of coats, retromer is recruited to membranes in part by the small GTPase, Rab7 (refs 14–19), which binds to VPS35 in the CSC subcomplex[16,18]. Members of the SNX family are peripheral membrane proteins and VPS35 has been shown to bind SNX3 simultaneously with Rab7, leading to cooperative, bivalent membrane recruitment of retromer[19]. By further analogy to other coats, the CSC also interacts with the Rab GAP, TBC1d5, in yeast two-hybrid and immunoprecipition experiments[20,21]. Overexpression of TBC1d5 leads to displacement of CSC from endosomes, an activity that has been ascribed to its GAP activity toward Rab7 (ref. 21). These observations suggest that coating and uncoating of membranes by retromer may follow the same logic as in the better known COPI and COPII coats, with GTPase recruitment being temporally opposed by a coat-associated GAP.

Despite these advances, several important questions remain to be answered about the retromer recruitment cycle: (1) does TBC1d5 directly interact with the retromer CSC, and if so, how? (2) How does the interaction of TBC1d5 with retromer affect GAP activity towards Rab7? (3) What are the cellular consequences of the TBC1d5-CSC interaction? To address these questions we have performed a combination of structural, biochemical and cellular studies. We show that the retromer CSC directly binds TBC1d5 with nanomolar affinity, through contacts to VSP29 and probably VPS35. The crystal structure of a VPS29-TBC1d5 complex reveals that a hydrophobic pocket of the retromer subunit binds an extended loop from the TBC GAP domain. The TBC1d5/CSC complex has higher GAP activity and affinity towards Rab7 than does TBC1d5 or CSC alone. Finally, we show that the interaction between TBC1d5 and retromer is critical for endosomal recycling of known retromer cargoes. Our study illustrates how TBC1d5 interacts with retromer and demonstrates that the interaction is important for retromer-mediated receptor trafficking.

## Results

**TBC1d5 directly interacts with the retromer CSC**. TBC1d5 is implicated in interacting with CSC and in regulating its endosomal localization by prompting hydrolysis of GTP on Rab7. However, it is unclear whether the interactions are direct and whether TBC1d5 can contact both Rab7 and CSC simultaneously. The amino terminus of TBC1d5 harbours a TBC domain, which probably functions as a GAP for the Rab7 GTPase (see below). The carboxy terminus is predicted to be largely disordered, except for a coiled-coil around amino acids 580–650. To test whether the binding is direct, we performed glutathione S-transferase (GST) pull-down assays using purified GST–TBC1d5 and CSC (Fig. 1a,b). Full-length GST–TBC1d5 could efficiently pull down the CSC. Further truncation studies indicated that two constructs encompassing the TBC (GAP) domain of TBC1d5 (TBC1d5$^{F1}$ and TBC1d5$^{TBC}$), but not a construct that does not include it (TBC1d5$^{F2}$) could retain the CSC, suggesting that the TBC domain is necessary and sufficient for the interaction. These data with purified proteins are consistent with a previous study showing that the first 548 amino acids of TBC1d5 are sufficient to immunoprecipitate CSC from cell lysates[20]. An isothermal titration calorimetry (ITC) titration of TBC1d5$^{TBC}$ into CSC yielded a $K_D$ value of $220 \pm 10$ nM, with 1:1 stoichiometry (Fig. 1c). This affinity is comparable to that measured between VPS35 and VPS29 under similar conditions ($K_D = 200$ nM)[22]. Finally, purified TBC1d5$^{TBC}$ and CSC co-eluted during gel filtration chromatography (Supplementary Fig. 1). In contrast, several established interaction partners of the CSC, including the WASH complex subunit FAM21 and SNX3, failed to co-elute with CSC, suggesting low affinity (Supplementary Fig. 1)[23–25]. Therefore, TBC1d5 distinguishes itself from other known binding partners of CSC by forming a direct, high-affinity interaction.

**Two unique insertions within TBC1d5 interact with CSC**. We next sought to determine how TBC1d5 interacts with the trimeric CSC assembly. The human genome encodes over 40 TBC domain-containing proteins. TBC1d5 is the only family member known to interact with the CSC and contains two unique insertions in its TBC domain: Insertion 1 (Ins1, aa 127–153) and Insertion 2 (Ins2, aa 263–293; Supplementary Fig. 2). We generated mutant proteins in which residues from either insertion were replaced by a (GGS)$_4$ linker (TBC1d5$^{TBC}$-ΔIns1 and -ΔIns2) and analysed them *in vitro* and in cells. According to several known crystal structures of the TBC domain[26], the linkers are sufficiently long to span between the cut points used in the truncated proteins. *In vitro*, we found by ITC that TBC1d5$^{TBC}$-ΔIns2 binds to CSC ∼9-fold weaker than TBC1d5$^{TBC}$ wild type (WT; $K_D = 2.0 \pm 0.3\,\mu$M versus $0.22 \pm 0.01\,\mu$M), whereas the affinity between TBC1d5$^{TBC}$-ΔIns1 and CSC was too weak to be determined accurately ($K_D > 10\,\mu$M; Fig. 1d). Similarly, both

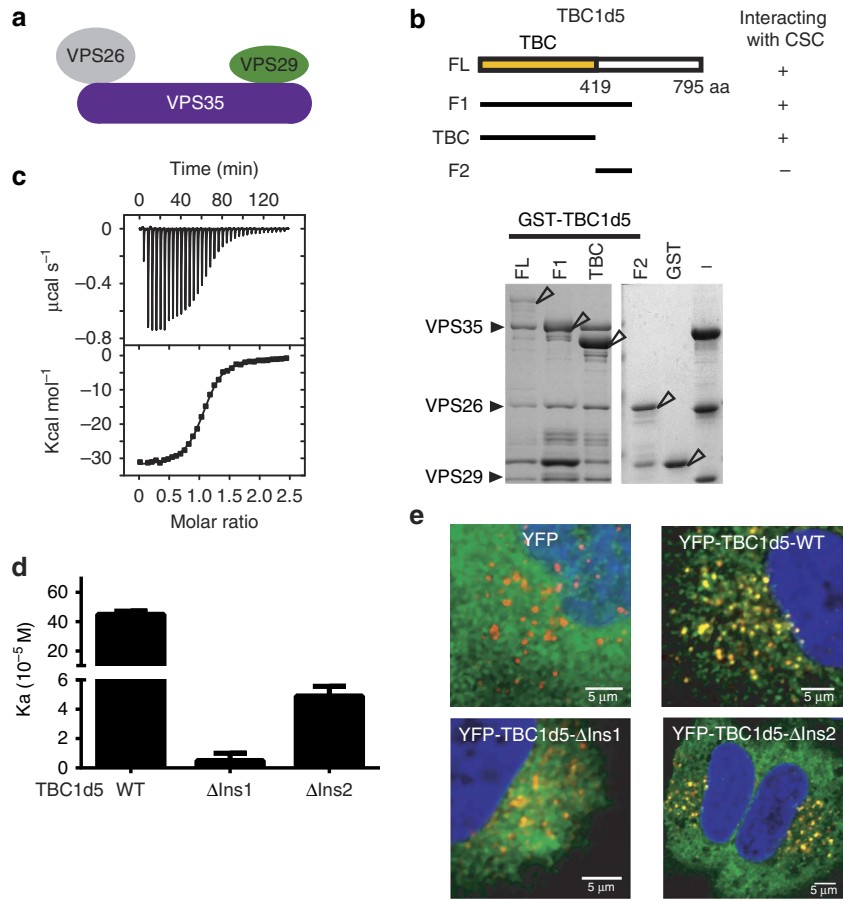

**Figure 1 | TBC1d5 forms a tight complex with the retromer CSC. (a)** Schematic representation of retromer CSC. **(b)** GST–TBC1d5 pull down of purified retromer CSC. Shown is the TBC1d5 constructs used (top) and a Coomassie blue stained SDS–PAGE gel of bound samples. Open triangles indicate GST–TBC1d5 or GST proteins. **(c)** ITC of TBC1d5$^{TBC}$ titrated into CSC in buffer containing 20 mM Tris-HCl pH 8.0, 200 mM NaCl and 5 mM βME at 20 °C. Top and bottom panels show raw and integrated heat from injections, respectively. The black curve in the bottom panel represents a fit of the integrated data to a single-site binding model. **(d)** Affinity between CSC and TBC1d5$^{TBC}$ WT, ΔIns1 or ΔIns2 determined by ITC. Association constant ($K_a$) are shown together with errors from data fitting. **(e)** Subcellular localization of TBC1d5 WT, ΔIns1 and ΔIns2. Hela cells were transfected with yellow fluorescent protein (YFP) or various YFP–TBC1d5 (green) and then fixed and labelled with anti-VPS35 antibodies (red). The larger images from which these were captured are shown in Supplementary Fig. 3B. Scale bar, 5 μm.

proteins were defective in co-purifying with CSC in gel filtration chromatography, with TBC1d5$^{TBC}$-ΔIns1 showing greater impairment (Supplementary Fig. 3A). In cells we found that although YFP–TBC1d5-WT strongly co-localized with VPS35, TBC1d5-ΔIns1 was primarily cytosolic and TBC1d5-ΔIns2 was found associated with small VPS35 punctae (Fig. 1e and Supplementary Fig. 3B). Further, TBC1d5-ΔIns1 and TBC1d5-ΔIns1 + 2 did not co-immunoprecipitate with VPS35, whereas WT TBC1d5 and, in some instances, TBC1d5-ΔIns2 did (Supplementary Fig. 3C). Thus, TBC1d5 uses both of its insertions to bind CSC, but Ins1 is more important for the interaction (Fig. 1d,e and Supplementary Fig. 3A–C).

We next sought to determine which protein(s) within the CSC are involved in the interaction with TBC1d5. The trimeric assembly of CSC is organized around VPS35, which consists of 17 HEAT repeats and forms an extended α-helical solenoid[27]. VPS26 and VPS29 bind to the N terminus and C terminus of VPS35, respectively. We found using gel filtration chromatography and ITC that both VPS35 and VPS29, but not VPS26, are required for the interaction with TBC1d5 (Supplementary Fig. 3D–F). Finally, by ITC, an Ins1 peptide could bind CSC and VPS29 with similar affinities, albeit at least one order of magnitude lower than for TBC1d5$^{TBC}$ (Supplementary Fig. 3D,G). Therefore, the interaction between

TBC1d5 and CSC depends primarily on Ins1 of TBC1d5, as well as both VPS35 and VPS29 of CSC.

**Crystal structure of VPS29/TBC1d5-Ins1.** To understand how VPS29 interacts with TBC1d5 in more detail, we determined a 1.5 Å crystal structure of the complex of full-length human VPS29 and the Ins1 peptide (residues 132–158 and a C-terminal His$_6$ tag; Fig. 2a and Table 1). In the complex with Ins1, the structure of VPS29 is very similar to previously reported structures of isolated VPS29 (root mean squared deviation = 1.2 Å for 170 aa) and VPS29 bound to VPS35 (root mean squared deviation = 0.7 Å for 182 aa)[27–29]. In all three structures, VPS29 adopts a phophoesterase fold, with a central β-sandwich flanked by two α-helices (α1 and α2) on one side and a third (α3) on the opposite side.

Previous structural studies revealed two conserved hydrophobic patches located on opposite surfaces of VPS29. One hydrophobic patch around residues V90 and I91 forms the primary contact site for VPS35 (see Fig. 2b). The second patch is formed by multiple conserved residues including L2, L25, V27, L152 and Y165, which form a surface-exposed hydrophobic cavity. Ins1 binds in a surface groove on VPS29 that centres around this cavity, positioning TBC1d5 and VPS35 on opposite

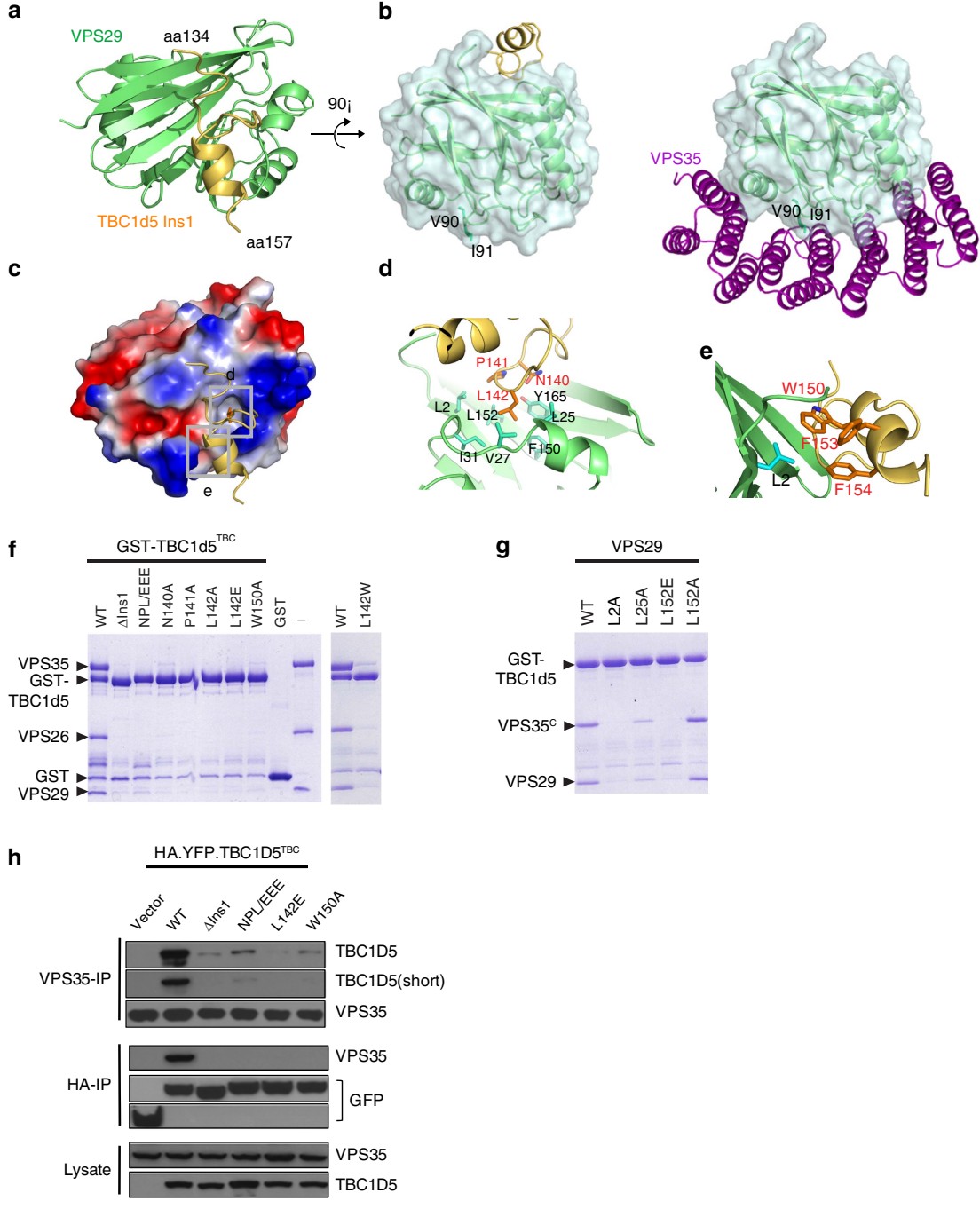

**Figure 2 | Interaction between VPS29 and TBC1d5.** (**a**) Ribbon diagram of the VPS29/TBC1d5-Ins1 complex (VPS29: green; Ins1; yellow). (**b**) Structural comparison of VPS29/TBC1d5-Ins1 and VPS29/VPS35[C]. VPS29 is shown as same orientation in two structures and is 90° rotation about a horizontal axis from **a**. (VPS35[C]: magenta). (**c**) GRASP surface charge distribution (blue for positive, red for negative, white for neutral). The Ins1 is shown as stick models. The structure is shown as same orientation as **a**. (**d,e**) VPS29-TBC1d5-Ins1 interactions in detail. Critical VPS29 and TBC1d5-Ins1 residues are shown in stick modes. (**f,g**) Coomassie blue stained SDS–PAGE gels of eluted proteins are shown. In **f**, immobilized GST–TBC1d5[TBC] WT but not mutants (ΔIns1, NPL/EEE, N140A, P141A, L142A, L142E, L142W and W150A) selectively retained CSC. In **g**, mutation on VPS29 surface residues had various effects on the interaction between GST–TBC1d5[TBC] and VPS35[C]/VPS29. (**h**) TBC1d5[TBC] WT, but not ΔIns1, NPL/EEE, L142E and W150A, co-immunoprecipitated with VPS35. Hela cells were transfected with yellow fluorescent protein (YFP) or various YFP–TBC1d5 (green) and immunoprecipitate with anti-VPS35 or anti-HA antibodies. Lysates were immunoblotted as a control.

surfaces of VPS29 (Fig. 2b,c). Residues 134–157 are represented by clear electron density in the structure (Supplementary Fig. 4). The Ins1 peptide can be divided into three distinct regions based on secondary structure (Fig. 2c): the N-terminal part (residues 134–137) forms an unstructured coil, the middle region (aa 138–148) makes multiple sharp turns on the VPS29 surface

and the C-terminal segment (residues 148–157) forms 1.5 turns of α-helix.

Recognition of Ins1 by VPS29 is predominately achieved through hydrophobic interactions (Fig. 2d,e). L142 of TBC1D5 lies in the centre of the Ins1-VPS29 interface. The side chain of L142 inserts into a deep and narrow cavity formed by L2, L25,

V27, I31, L152 and Y165 of VPS29. Immediately preceding L142, P141 makes van der Waals contacts with L152, Y163 and Y165 of VPS29, and also helps to make a sharp turn, to allow L142 to contact the hydrophobic residues from VPS29. Two residues preceding L142, N140 forms a hydrogen bond with the side chain

of Y165 in addition to making van der Waals contacts with Y163 and Y165 of VPS29. The residues immediately following L142 make a few sharp turns within a limited distance (only 6.3 Å between Cα of S143 and S148 versus 17.5 Å for an extended structure). This region is dominated with small amino acids (SQDEGS) and their size may be necessary to accommodate the space. In addition, from the C-terminal region of Ins1, W150, F154 and F156 of TBC1d5 form a hydrophobic cluster on one face of the helix and pack against L2 of VPS29 (Fig. 2e).

To verify our structure, we purified a series of TBC1d5$^{TBC}$ mutant proteins encompassing single or triple substitutions for residues that contact VPS29 and tested their ability to pull down the CSC (Fig. 2f). All these mutants abolished or nearly abolished binding to the CSC. Most notably, conversion of L142 to a small (L142A) or large (L142W) hydrophobic residue, or a charged residue (L142E) significantly disrupted its interaction with CSC, suggesting that both size and charge are important to contact VPS29. We also made a series of mutations on VPS29 residues that interact with TBC1d5 Ins1 (Fig. 2g). These bound the C terminus of VPS35 (VPS35$^C$) normally in gel filtration chromatography (not shown). However, the amount of VPS35$^C$/VPS29-L2A, -L25A, or -L152E retained by GST–TBC1d5$^{TBC}$ was significantly reduced relative to that of VPS35$^C$/VPS29-WT. In contrast, the amount of VPS35$^C$/VPS29-L152A was not affected. Similar to our results with purified proteins, analogous mutants in Ins1 largely abolished the co-immunoprecipitation of TBC1d5 and VPS35 (Fig. 2h). Finally, the sequence conservation among VPS29 and TBC1d5 orthologues strongly suggests that the interactions between CSC and TBC1d5 were conserved during evolution (Supplementary Fig. 5).

In addition to the Ins1-VPS29 complex, we also determined a 3.8 Å crystal structure of the complex between TBC1d5$^{TBC}$, VPS29 and the C terminus of VPS35 (VPS35$^C$) (Supplementary Fig. 6). The three proteins form a sandwich, with VPS29 flanked

**Table 1 | Crystallography data collection and refinement statistics.**

| | |
|---|---|
| *Data collection* | |
| Space group | P2$_1$2$_1$2$_1$ |
| Cell dimensions | |
| $a$, $b$, $c$ (Å) | 43.88, 63.83, 78.01 |
| $\alpha$, $\beta$, $\gamma$ (°) | 90, 90, 90 |
| Resolution (Å) | 50-1.50 (1.53-1.50)* |
| $R_{merge}$ | 0.066 (0.91) |
| $I/\sigma I$ | 38 (1.2) |
| Completeness (%) | 98.5 (84.8) |
| Redundancy | 5.7 (4.5) |
| | |
| *Refinement* | |
| Resolution (Å) | 50-1.5 |
| No. reflections | 35,136 |
| $R_{work}/R_{free}$ | 16.6%/19.2% |
| No. non-hydrogen atoms | |
| Protein | 1,670 |
| Water | 140 |
| B-factors | |
| Protein | 31.2 |
| Water | 38.5 |
| Root mean squared deviations | |
| Bond lengths (Å) | 0.012 |
| Bond angles (°) | 1.57 |

*Values in parentheses are for the highest resolution shell.

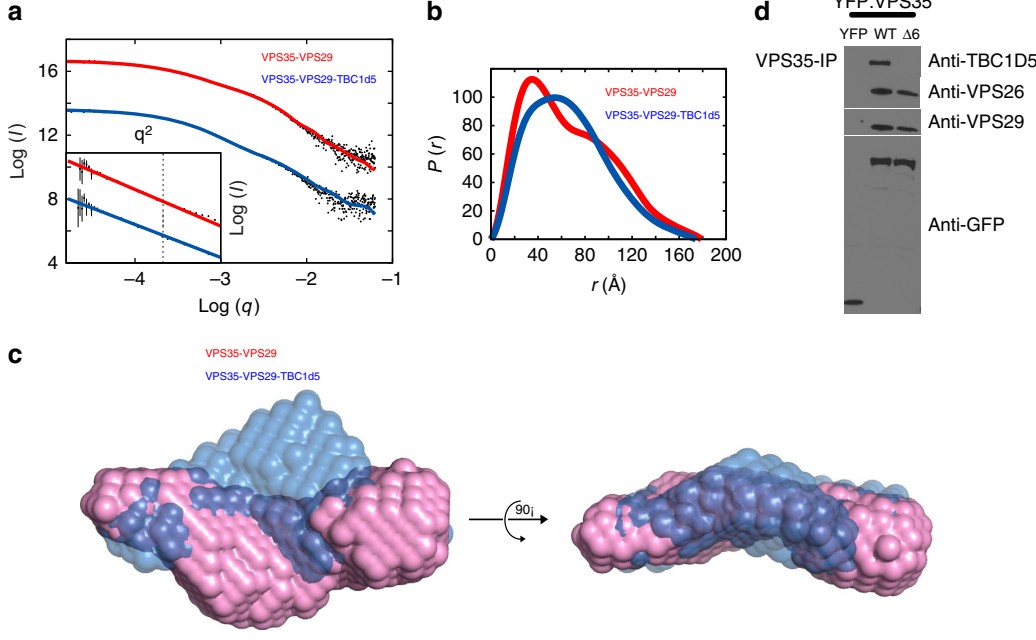

**Figure 3 | Interaction between VPS35 and TBC1d5. (a)** SAXS scattering curves. The VPS35/VPS29 binary and VPS35/VPS29/TBC1d5$^{TBC}$ ternary scattering curves are shown as black points. The CORAL rigid-body model fits are shown as red (Binary) and blue (Ternary) lines. Inset: the Guinier plot for each data set is shown with the linear fit coloured as in the figure. The Ternary curve is offset for clarity. **(b)** P(r) curves. The Binary (blue) and Ternary (red) pair–distance distribution functions as determined by *BayesApp*. **(c)** SAXS *ab initio* molecular shapes. Overlaid Dammif bead models, Binary (red) and Ternary (blue) are shown in two different views. **(d)** VPS35 WT, but not Δ6, co-immunoprecipitated with TBC1d5. Hela cells were transfected with yellow fluorescent protein (YFP) or various YFP–VPS35 and immunoprecipitate with anti-VPS35 antibodies. TBC1d5, VPS26, VPS29 and green fluorescent protein (GFP) were immunoblotted. See also Supplementary Fig. 8.

 5

by TBC1d5 and VPS35. TBC1d5 does not contact VPS35[C], consistent with our biochemical data that the N terminus of VPS35 is necessary to form a tight TBC1d5/CSC complex. TBC1d5 interacts with VPS29 only through a region corresponding to Ins1 (Supplementary Fig. 6B–E). Although the electron density in this region is weak, within the resolution limits of the larger structure, Ins1 and VPS29 have the same conformations and make the same contacts in the Ins1-VPS29 and TBC1d5[TBC]-VPS29-VPS35[C] complexes. Thus, the global orientation of TBC1d5 relative to VPS29 and VPS35[C] may be different from that observed in the crystal, particularly in the presence of full-length VPS35, which probably makes additional contacts to Ins2 of the GAP domain (see below).

**Interaction between TBC1d5 and VPS35.** As our biochemical and cell biological data suggest that, in addition to VPS29, VPS35 also contributes to the tight complex with TBC1d5, we next sought to understand how TBC1d5 interacts with VPS35. Attempts to co-crystallize TBC1d5 with full-length VPS35 and VPS29 failed; therefore, we used small-angle X-ray scattering (SAXS) to determine how the VPS35/VPS29/TBC1d5[TBC] complex is assembled. SAXS data were collected for both the VPS35/VPS29 binary and the VPS35/VPS29/TBC1d5[TBC] ternary complex. Data for both the binary and ternary complexes gave good Guinier fits (Fig. 3a insert) and a dilution series indicated that our data were free from interparticle affects. The molecular weights, calculated from both the $I_o$ and the Porod volume agree very well with the known masses of these complexes (Supplementary Table 1). Furthermore, the radius of gyration ($R_G$) and maximum dimensions ($D_{max}$) for the binary complex are similar to a previous study[22]. The binary and ternary complexes have similar $R_G$ (52.4 Å for binary versus 50.7 Å for

ternary), as well as $D_{max}$ (179 versus 174 Å; Fig. 3b). This suggests that TBC binding does not cause a large conformation change in VPS35, and that it probably binds near the centre of mass of the elongated binary complex.

The Dammif *ab initio* molecular shapes for both complexes were similar, except for a bulge near the centre of the ternary bead model, indicating where TBC binds to VPS29 (Fig. 3c). These *ab initio* models suggest that the VPS35 heat repeat continues its slight curvature around VPS29, as in the VPS35[C]/VPS29 crystal structure[27]. The bulge for the TBC molecule would not be inconsistent with a contact to the VPS35[N] region in addition to binding VPS29. Owing to the lack of high-resolution structures for TBC1d5 and full-length VPS35, we did not perform rigid body modelling.

HEAT repeat 6 within VPS35 is highly conserved among species and it was previously shown that deletion of this element (to give VPS35-Δ6) impacted interaction of the protein with Rab7 and its endosomal accumulation, without altering its ability to bind VPS26 and VPS29 (ref. 16). We tested whether Repeat 6 of VPS35 was necessary to bind TBC1d5. Indeed, in contrast to WT VPS35, VPS35-Δ6 could not immunoprecipitate TBC1d5 (Fig. 3d). As expected, VPS35-Δ6 could interact with VPS26 and VPS29. Next, we sought to identify specific residues within R6 of VPS35 that could interact with TBC1d5. Three hydrophobic residues (I283P284F285) from Ins2 of TBC1d5 are important for the interaction with CSC, because their mutation to glutamic acid (TBC1d5[TBC]-IPF/EEE) disrupted the chromatographic co-purification with CSC, similar to deletion of Ins2 (Supplementary Fig. 8). We therefore focused on testing hydrophobic resides within R6 of VPS35. Mutation of several residues (Y261L262M263/AAA, I266I267/AA) in R6 had little effect on interactions with TBC1d5 (Supplementary Fig. 7). However, mutation of two hydrophobic residues, V269F270,

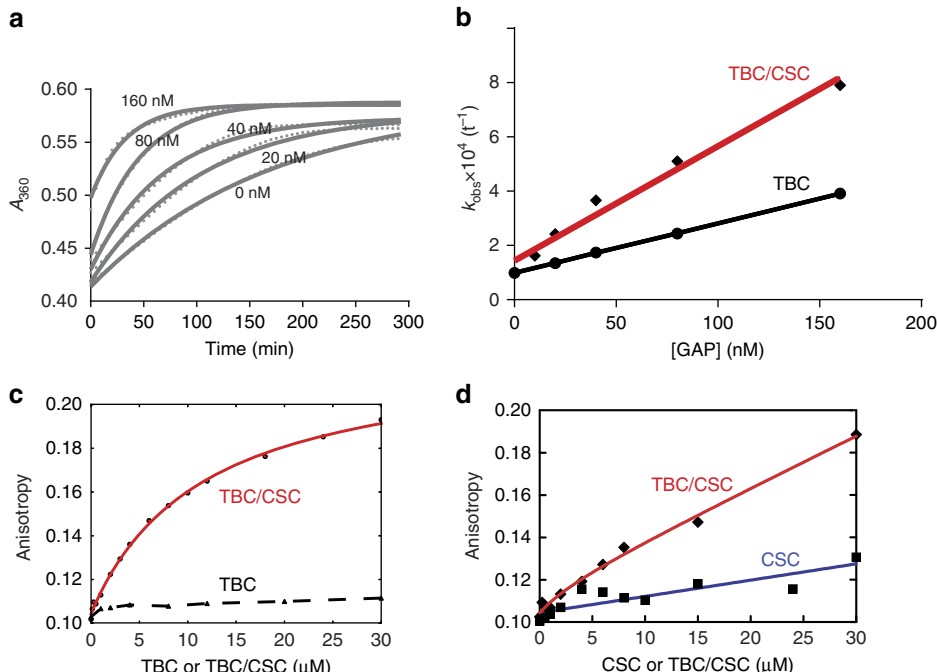

**Figure 4 | CSC promotes the GAP activity of TBC1d5 and its interaction with Rab7. (a)** Kinetics of GTP hydrolysis for Rab7 in the absence and presence of the indicated concentrations of TBC1d5[TBC]. The absorbance at 360 nm reports the amount of a conjugate of GTP hydrolysis product, Pi. Solid lines represent a fit of experiment data to the pseudo-first-order Michaelis–Menten equation. **(b)** Calculated $k_{obs}$ for TBC1d5[TBC] or TBC1d5[TBC]/CSC as a function of of Rab7 concentration. Solid lines represent a linear fit between $k_{obs}$ and Rab7 concentration, giving rise to catalytic efficiency ($k_{cat}/K_m$, slope) and intrinsic rate constant (y intercept). **(c)** Fluorescence anisotropy direct binding assay, in which TBC1d5[TBC] or TBC1d5[TBC]/CSC was added to Alexa Fluor 488-labelled Rab7-GMPPNP. Solid lines represent a fit with changes in fluorescence anisotropy. **(d)** Fluorescence anisotropy direct binding assay, in which CSC or TBC1d5[TBC]/CSC was added to Alexa Fluor 488-labelled Rab7-GMPPNP.

to arginine (VPS35-V269F270/RR) abolished co-purification of VPS35 with TBC1d5 (Supplementary Fig. 7) without affecting interactions with VPS26 and VPS29. As deletion or mutation of R6 weakens binding to TBC1d5, this repeat or proximal elements of VPS35 that are affected by these perturbations, appears to contribute to the CSC-TBC1d5 interaction.

**CSC increases the GAP activity of TBC1d5 towards Rab7**. One previous study showed that the TBC1d5 paralogue in *Caenorhabditis elegans*, RBG-3 (37% sequence identity in the TBC domain), selectively regulates Rab7 among a large panel of Rab GTPases[30]. We therefore tested the ability of TBC1d5[TBC] or TBC1d5[TBC]/CSC, to promote GTP hydrolysis by Rab7. We found that TBC1d5[TBC] can activate Rab7 GTP hydrolysis in a concentration-dependent manner (Fig. 4a). Furthermore, the catalytic efficiency ($k_{cat}/K_m$) of TBC1d5[TBC] for Rab7 is $1,800\,M^{-1}\,s^{-1}$, comparable with the reported value for its *C. elegans* paralogue (Fig. 4b)[30]. The TBC1d5[TBC]/CSC complex displayed about 2.5-fold higher activity than TBC1d5[TBC] alone (Fig. 4b). In contrast with TBC1d5[TBC] or the TBC1d5[TBC]/CSC complex, no GAP activity was detected for CSC alone, even at 320 nM concentration (Supplementary Fig. 8A).

Using GST pull-down assays, we found that Rab7 retained the TBC1d5[TBC]/CSC complex, but not CSC alone containing VPS35-WT or VPS35-V269F270/RR, in a GMPPNP-dependent manner (Supplementary Fig. 8B). We furthur employed fluorescence anisotropy assays to quantify the interaction between Rab7 and TBC1d5 or TBC1d5/CSC (Fig. 4c and Supplementary Fig. 8C,D). Titration of TBC1d5[TBC]/CSC, but not TBC1d5[TBC] alone, into AlexaFluor488-labelled Rab7-GMPPNP caused a substantial increase in fluorescence anisotropy (Fig. 4c). Addition of excess unlabelled WT Rab7-GMPPNP competed for binding (Supplementary Fig. 8C). Fitting the direct binding and competition data to single site-binding isotherms indicates that TBC1d5[TBC]/CSC binds to labelled and unlabelled Rab7-GMPPNP with $K_D$ values of $11.5 \pm 0.8$ and $36 \pm 7\,\mu M$, respectively. Addition of a ninefold excess TBC1d5[TBC] protein could not compete TBC1d5[TBC]/CSC away from Rab7 (Supplementary Fig. 8D). Finally, fluorescence anisotropy assay suggests that CSC does not interact measurably with Rab7-GMPPNP. It is surprising that there is only a modest difference between the activity of TBC and TBC/CSC, but an apparently large difference in affinity. This could result from the nucleotide difference: GTP in the activity assay and GMPPNP in binding assay. Alternatively, it is possible that the changes in anisotropy reflect binding of Rab7 to a site on Vps35 in the CSC, whose affinity is enhanced by the presence of TBC1d5. Regardless of mechanism, the data indicate that the TBC1d5[TBC]/CSC complex binds Rab7 more tightly than either TBC1d5[TBC] or CSC alone.

**TBC1d5 is required for retromer-mediated receptor recycling**. In the COPII system, Sec23 is a GAP for Sar1 and the Sar1-Sec23 interaction leads to the recruitment of COPII to endoplasmic reticulum membranes. We asked whether TBC1d5, the GAP for Rab7, could play an analogous role in the membrane recruitment of retromer CSC. To test this, we generated TBC1d5 knockout (KO) HeLa cells using CRISPR/Cas9 gene-editing technology (Fig. 5a). Importantly, deletion of TBC1d5 did not affect the total levels of VPS35 (Fig. 5a). To assess the impact of TBC1d5 loss on VPS35 recruitment to endosomes, TBC1d5 KO cells were mixed with control HeLa cells at a ratio of 1:1 and the cells were simultaneously stained with TBC1d5, VPS35 and EEA1. As shown in Fig. 5b, loss of TBC1d5 does not impact the recruitment of VPS35 to endosomes.

Next, we asked whether the loss of TBC1d5 might have an impact on receptor recycling. Integrin α5β1 is a known retromer cargo that recycles through the endosomal system back to the plasma membrane[31,32]. To examine the impact of TBC1d5 loss on α5 (CD49e) trafficking, we performed an assay in which we fed anti-CD49e antibody to control and TBC1d5 KO HeLa cells mixed 1:1. In TBC1d5[+] cells, CD49e efficiently recycles and only shows small internal puncta of CD49e that are associated with retromer (Fig. 5c). However, in cells where TBC1d5 is deleted, CD49e becomes trapped in retromer-coated endosomes (Fig. 5c and Supplementary Fig. 9A), demonstrating a more than threefold higher mean fluorescence intensity of endosomal-CD49e in TBC1d5 KO cells compared with control cells (Fig. 5d). Thus, TBC1d5 is clearly involved in endosome-to-plasma membrane receptor recycling from this sorting domain.

To examine the role of TBC1d5 in endosome-to-Golgi retrieval, we assessed the localization of CI-M6PR, a well-established retromer cargo that recycles between endosome and TGN[33]. Although cation-independent mannose 6-phosphate receptor (CI-MPR) is clustered in parental cells, consistent with the localization on TGN, TBC1d5 KO led to dramatically increased CI-MPR dispersal (Fig. 5e,f) and punctate localization in the cytosol (Supplementary Fig. 9B). Taken together, these data suggest that recruitment of TBC1d5 to the CSC is also required for endosome-to-Golgi retrieval of this retromer cargo.

**Discussion**
We present multiple lines of evidence suggesting that TBC1d5 is an important regulator for the retromer coat: (1) TBC1d5 forms a stoichiometric complex with CSC and the complex is stable under many different conditions. The interaction between TBC1d5 and CSC is as tight as those among different subunits of CSC. Numerous proteins are known to interact with CSC, including SNX proteins, FAM21, EHD1 and the recently described adaptor protein VARP[12,34,35]. The biochemical interaction between CSC and many binding partners has not been extensively characterized. However, among a few that have been studied, the affinity between TBC1d5 and CSC is the highest by one to two orders of magnitude[35,36]. Whereas a number of proteins could co-immunoprecipitate with VPS29 in mild conditions, TBC1d5 is the only protein, in addition to VPS35 and VPS26, that co-immunoprecipitates with VPS29 in the presence of 1% Triton X-100 (ref. 37). (2) TBC1d5 associates with the CSC through a binary interaction involving both VPS35 and VPS29. The interaction with TBC1d5 covers a large hydrophobic surface on VPS29, which is otherwise solvent exposed. Likewise, the association between TBC1d5 and VPS35 is likely to be mediated through hydrophobic interactions. The nature of these interactions also indicates TBC1d5 is an important binding partner of the complex. (3) We found that TBC1d5 is indispensable for both retromer-mediated trafficking routes: endosome-to-Golgi and endosome-to-plasma membrane. (4) We also found that CSC could promote the GAP activity of TBC1d5. Binding to CSC probbaly causes a conformational change in TBC1d5 that increases both its affinity for Rab7 and its GAP activity. Alternatively, CSC may harbour a site that weakly binds to Rab7.

The interaction with TBC1d5 is likely to be a general theme for the retromer coat across eukaryotes. Similar to other retromer components, TBC1d5 orthologues are widely distributed among eukaryotes (Supplementary Fig. 10A). Our sequence alignment indicates that key residues important for the TBC1d5–VPS29 interaction are evolutionarily conserved. Previously, studies in multiple model organisms, including *Entamoeba histolytica* and *Arabidopsis thaliana*, have established the importance of Rab7 for the membrane assembly of the retromer CSC[14,17]. Our studies

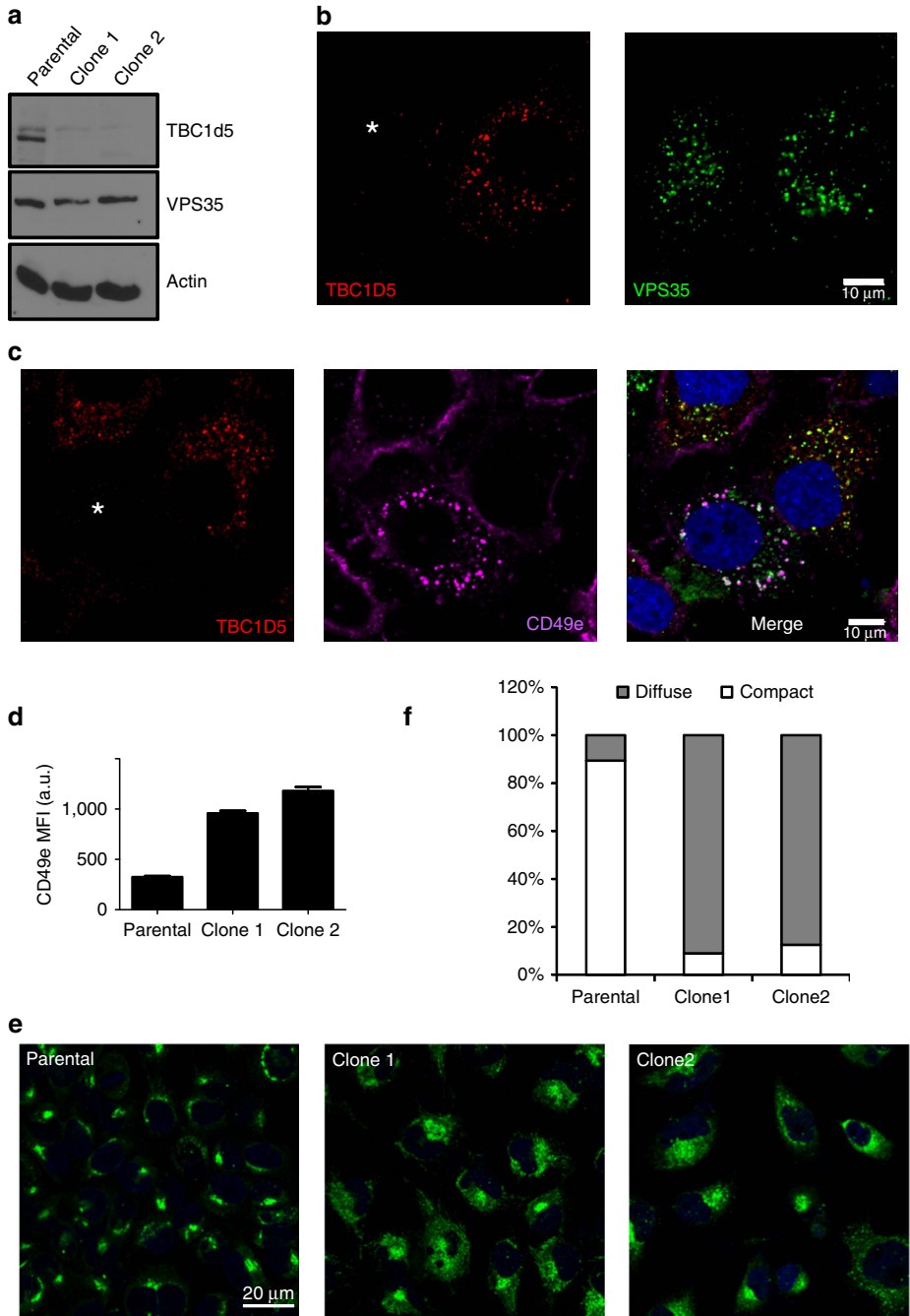

**Figure 5 | TBC1d5 is required for retromer-mediated trafficking but not endosomal recruitment of VPS35.** (**a**) Levels of TBC1d5, VPS35 and actin expression in TBC1d5 KO (clone 1 and 2) HeLa cell line and parental cells determined by immunoblotting. (**b**) Parental and clone 1 HeLa cells were mixed with 1:1, fixed and stained for TBC1d5 (red) and VPS35 (green). Asterisk represents cell that is devoid of TBC1d5. Scale bar, 10 μm. (**c**) Parental and clone 1 HeLa cells were mixed with 1:1 and incubated with anti-CD49e antibody. The cells were then fixed and stained for TBC1d5 (red), VPS35 (green) and CD49e (magenta). Asterisk represents cell that is devoid of TBC1d5. Scale bar, 10 μm. (**d**) The average mean fluorescence intensity (MFI) of retromer-localized CD49e in parental, clone 1 or clone 2 HeLa cells, with error bar representing s.e.m. Cells were fed anti-CD49e antibody for 1 h, then fixed and stained for internalized CD49e and VPS35. MFI was measured using Zen2009 software. One hundred cells from each group were used for comparison. (**e**) TBC1d5 is important for CI-MPR dispersal. Parental, clone 1 or clone 2 Hela cells were fixed and stained with anti-CI-MPR. Scale bar, 20 μm. (**f**) Quantification of **e**. One hundred cells from each group were scored for either compact or dispersed CI-MPR distributions.

indicate that such events probably depend on their TBC1d5 orthologues. *Saccharomyces cerevisiae* lacks a TBC1d5 homologue despite having a functional retromer system. It seems likely to be that the retromer CSC in *S. cerevisiae* has evolved to associate with a distinct class of Rab7 GAP, or perhaps to possess GAP activity in its Rab7 interaction region, more akin to the Sec23-Sar1 system.

Our study, together with previous published data, begins to suggest a model by which retromer-mediated receptor recycling could be regulated (Supplementary Fig. 10B). On endosomal loading of Rab7-GTP and membrane association of SNX3 (ref. 19), CSC is recruited through its interaction with Rab7, bringing TBC1d5 with it. The TBC1d5/CSC complex subsequently promotes GTP hydrolysis of Rab7, which triggers

the disassembly and release of CSC from membranes, which is required for receptor recycling. As in the COPI and COPII systems, coupling the coat to a GAP that inactivates the recruiting GTPase probably provides important timing information to the coating/uncoating process. Further work will be necessary to address how TBC1d5 functions together with other regulators, such as SNX, and the WASH complex to regulate retromer assembly and turnover on endosomal membranes.

## Methods

**Antibodies and plasmids.** Reagents were from Sigma, unless specified. Anti-HA Affinity Matrix and anti-HA-HRP were from Roche. Fluorescently conjugated secondary reagents were from Molecular Probes. DNA constructs and antibodies used in this study are listed in Supplementary Tables 2 and 3, respectively.

**Cell culture.** HeLa and HEK293T cells (obtained from ATCC) were grown in RPMI 1640 medium with 5% fetal bovine serum, 5% FCS and 4 mM L-glutamine, and were transfected using electroporation as described[24]. For immunoprecipitation experiments, cells were lysed in NP-40 lysis buffer (25 mM HEPES pH 7.9, 50 mM NaCl, 1 mM EDTA, 0.5 mM CaCl₂, 0.5% NP-40, 1 mM phenylmethyl sulfonyl fluoride, 10 μg ml⁻¹ leupeptin and 5 μg ml⁻¹ aprotinin) and 500–1,000 μg was prepared and analysed by immunoblotting as described[24]. TBC1d5-CRISPR cell lines were generated through stable expression of a targeting guide RNA and Cas9, using the pLENTI-CRISPR vector. KO clones were isolated through serial dilutions and screened for TBC1d5 expression by immunoblotting.

**Immunofluorescence and CI-MPR sorting assay.** HeLa cells were grown directly on coverslips, fixed in 4% paraformaldehyde and prepared for immunofluorescence as described[24]. Images were obtained with an LSM-710 laser scanning confocal microscope (Carl Zeiss). CI-MPR soring assays were performed as previous studies[24]. For anti-CD49e feeding assay, 1 μg of antibody was diluted into 100 μl of serum-free medium and cells grown on coverslips were incubated for 60 min at 37 °C. Cells were subsequently fixed and prepared for immunofluorescence as described above. Zen2009 software was used to measure the mean fluorescent intensity of internalized CD49e that was associated with VPS35 punctae in HeLa and TBC1d5 KO cells (100 cells each).

**Protein purification.** Purification of retromer CSC and subcomplexes were described previously[36]. Briefly, VPS35, VPS29 and VPS26 were individually expressed in E. coli BL21(DE3)-T1R (Sigma) and cell pellets were mixed and co-lysed by high-pressure homogenization. The complex was first purified by GST-fusion on the N terminus of VPS35. The GST-fusion was cleaved and further purified by Source Q ion exchange (GE Healthcare) and gel filtration (Superdex 200, GE Healthcare) chromatographies. The CSC can also be purified from bacteria expressing a polycistronic vector, expressing all three subunits, with a substantial lower yield. For the VPS35ᶜ/VPS29 complex, plasmids encoding VPS35ᶜ and VPS29 were co-transformed into E. coli BL21(DE3)-T1R and expressed at 16 °C overnight.

Various TBC1d5 fragments were inserted in a modified pGEX vector, resulting in a N-terminal GST tag following by a tobacco etch virus (TEV) protease site. The proteins were expressed in E. coli BL21(DE3)-T1R at 25 °C overnight or 30 °C for 6 h. After expression, the cells were lysed by high-pressure homogenization and the cleared lysate was subjected to glutathione-sepharose 4B affinity (GE Healthcare) chromatography. The bound proteins were eluted with reduced glutathione and cleaved with TEV protease. The cleaved proteins were further purified by ion exchange and gel filtration chromatographies. TBC1d5ᵀᴮᶜ was also expressed with a His-SUMO fusion and the protein was purified using Ni-NTA agarose beads (Qiagen) and a Source Q15 column. All TBC1d5ᵀᴮᶜ mutant proteins have similar yield and chromatographic behaviours similar to the WT protein. All mutants are monomeric by gel filtration chromatography.

**GST pull-down experiments.** GST pull-down experiments were performed as previous studies[36]. The mixture contained 700 pmol of GST or different GST–TBC1d5 proteins, and 1,500 pmol of purified retromer CSC for TBC1d5 pull-down. The proteins were mixed with glutathione sepharose 4B resin in 1 ml of pull-down buffer (20 mM Tris pH 8.0, 100 mM NaCl, 5% (w/v) glycerol, 5 mM β-mercaptoethanol (βME)). After gentle mixing at 4 °C for 30 min, the resin was washed four times with 1 ml of pull-down buffer, followed by elution with reduced glutathione. Eluted proteins were resolved by SDS–PAGE and visualized with Coomassie blue.

**Crystallography.** The TBC1d5 Ins1 peptide (aa 132–158) was expressed with an N-terminal TEV-cleavable maltose binding protein-tag and a C-terminal His6 tag. Following cell lysis, the fusion was affinity purified using Amylose resin (NEB). Following elution, the peptide was cleaved from the fusion with TEV protease at 4 °C overnight. The peptide was subsequently purified by Ni-NTA and gel filtration chromatography (Superdex 75 10/300 GL, GE Healthcare).

VPS29 (0.5 mM) was mixed with threefold excess of TBC1d5 Ins1 peptide for crystallization. Diffraction quality crystals were obtained from hanging drops containing 2 μl protein and 2 μl of well solution (0.1 M Bis-Tris pH 6.0, 2% Tacsimate pH 6.0 and 15∼20% PEG3350). Crystals were transferred from crystallization solution with 20% (w/v) glycerol and flash frozen in liquid nitrogen. Diffraction data were collected at beamline 19-ID at the Advanced Proton Source, Argonne National Laboratory. The data were processed and scaled using HKL3000 (ref. 38). The structure was solved by molecular replacement using Phaser[39] with VPS29 as search models. The model was improved through iterative cycles of model building in Coot[40] and refinement in Phenix[41].

To crystallize the VPS35ᶜ/VPS29/TBC1d5 complex, a 1:1 molar mixture of VPS35ᶜ/VPS29 and TBC1d5ᵀᴮᶜ proteins were concentrated to 5∼7.5 mg ml⁻¹. Crystals could be easily obtained from hanging drops containing 2 μl protein and 2 μl of well solution (0.1 M Hepes buffer pH 7.0, 0.8∼0.9 M Na Malonate and 5% (w/v) glycerol); however, they only diffracted to ∼10 Å. These crystals were transferred to a dehydrating solution (well solution with 1.2 M Na Malonate) overnight, or after crystal growth the cover slips were transferred to reservoirs containing the same dehydrating solution. The dehydrated crystals could routinely diffract to <5 Å, with the best one to 3.8 Å. The crystals belong to the space group $p3_121$ ($a = b = 122$ Å and $c = 272$ Å). The structure was solved by molecular replacement using Phaser (McCoy et al., 2007) with VPS35ᶜ/VPS29 (pdb 2R17) and GYP1P (pdb 1FKM) as search models at the resolution range of 35–3.79 Å (118,555 observed reflections, 24,137 unique reflections, 99% completeness, $R$-merge = 0.086, $<I/\sigma \geq 14.7$). Each asymmetric unit contains two copies of VPS35ᶜ/VPS29/TBC1d5ᵀᴮᶜ. Whereas most residues of VPS35ᶜ and VPS29 have clear electron density, we could not build side chains for most residues from TBC1d5ᵀᴮᶜ, owing to poor density, and the final refinement gives rise to $R$-factor and $R$-free values of 0.36 and 0.39, respectively.

**Isothermal titration calorimetry.** ITC experiments were conducted at 20 °C using a VP-ITC microcalorimeter (Microcal) as previously described[36]. Before each experiment, proteins were subjected to gel filtration chromatography or extensive dialysis, to exchange them into ITC buffer (20 mM Tris pH 8, 200 mM NaCl and 5 mM βME). In each experiment, TBC proteins (100∼200 μM) were injected into the sample cell containing retromer proteins (8–16 μM). Data were analysed with the Origin 7.0 software package (OriginLab) by fitting the 'one set of sites' model to the binding isotherm, yielding a single $K_d$ value for all binding events.

**Fluorescence anisotropy.** Fluorescence anisotropy experiments were performed at 20 °C using a PTI Fluorimeter (Photon Technology International)[42]. An AlexaFluor488-labelled Rab7 (aa 1–186-Cys and Cys143Ser) was used as the fluorescence anisotropy probe ($\lambda_{ex}$ = 495 nm, $\lambda_{em}$ = 519 nm). Rab7 (aa 1–186) contains three Cys residues. In initial labelling experiments with maleimide-AlexaFluor488, only Cys143 reacted with the dye as demonstrated by mass spectrometry and mutagenis. Therefore, Cys143 was mutated to Ser and a Cys residue was added to the C terminus of the protein. Mass spectrometry analyses of this protein after reaction with maleimide-AlexaFluor488 confirmed that only one site is modified, presumably the C-terminal cysteine residue. All protein samples were dialysed or desalted into FA buffer (20 mM Tris pH 8.0, 100 mM NaCl, 2 mM MgCl₂ and 5 mM βME) before use. G factor was calculated as $G = (I_{hv\ sample} - I_{hv\ buffer})/(I_{hh\ sample} - I_{hh\ buffer})$ and anisotropy was calculated using $r = (I_{vv} - G \times I_{vh})/(I_{vv} + 2G \times I_{vh})$. In direct binding measurement, TBC1d5 or TBC1d5/CSC proteins were titrated into 50 nM labelled Rab7-GMPPNP in 200 μl of reaction. The anisotropy data were fit using a single-site 1:1 binding model to obtain $K_D$. In competition measurements with cold Rab7, unlabelled Rab7-GMPPNP (aa 1–86) was titrated into reactions containing 8 μM TBC1d5/CSC and 50 nM labelled Rab7-GMPPNP. In the competition assay with TBC1d5, TBC1d5ᵀᴮᶜ was titrated into the same reactions. The dissociation constant $K_D$ was obtained using competitive binding one-site model integrated in GraphPad Prism version 5.00.

**GAP assay.** The GAP assays using the EnzChek Phosphate Assay Kit (Invitrogen) were carried out as previously described[26]. Briefly, Rab7 protein was incubated with 15-fold molar excess of GTP at room temperature for 30 min. Free nucleotides were removed by a desalting column pre-equilibrated with reaction buffer (20 mM HEPES pH 7.5 and 150 mM NaCl). Solution A containing GTP-preloaded Rab7 was mixed with solution B containing TBC1d5, enzyme and buffer. The final solution contained 20 mM HEPES pH 7.5, 150 mM NaCl, 10 mM MgCl₂, 200 μM 2-amino-6-mercapto-7-methylpurine riboside (MESG), 5 unit of purine nucleoside phosphorylase, 20 μM Rab7 and the indicated amount of TBC1d5. The absorbance at 360 nm was continuously monitored on a Varioscan Flash microplate reader (Thermo Scientific). Kinetics was determined following a previously described method.

**SAXS data collection.** SAXS data were collected on a Rigaku BioSAXS-1000, Kratky camera, with a 2.5 kW FRE+ source at the Sealy Center for Structural Biology, of the University of Texas Medical Branch, Galveston, TX. Thirty microlitres of sample or buffer were sealed in a quartz capillary cell at 10 °C. Data collection for each sample comprised a series of up to eight 30 min exposures for

the sample and fifteen 60 min exposures for the buffer, using the same cell. Sample curves were checked for the onset of radiation-induced changes by the cross $\chi^2$-metric[43] using the SAXNS-XCS server (http://xray.utmb.edu/SAXNS). The images were processed using SAXSLab (Rigaku) to produce averaged scattering curves for the sample and buffer. The averaged sample and buffer scattering curves were processed using SAXNS-ES (http://xray.utmb.edu/SAXNS). The resulting averaged and merged SAXS curves were used for subsequent analysis.

A dilution series for each sample was performed at $\sim 4$, 2, 1 and 0.6 mg ml$^{-1}$ concentrations. Guinier analysis indicated that the 0.6 mg ml$^{-1}$ samples were free of aggregates for both VPS35/VPS29 and VPS35/VPS29/TBC1d5$^{TBC}$. Therefore, the 0.6 mg ml$^{-1}$ samples, comprising of 10 (binary), and 8 (ternary) 30 min exposures, were used for later analysis.

**SAXS data analysis and *ab initio* model generation.** SAXS analysis started with an estimation of the radius of gyration (Rg) and Io using SAXSlab and Primus[44], which was also used to generate the GNOM file for DAMMIF *ab initio* molecular shape generation. Using saxns_dammif (http://xray.utmb.edu/SAXNS), 15 Dammif bead models were generated, then averaged and filtered with Damaver[45]. The resulting consensus model was then aligned and, if necessary, inverted to fit its respective rigid-body model using Supcomb[46]. Inversion of the envelope is sometimes necessary, because the SAXS curve is invariant under inversion and therefore its correct hand cannot be determined *ab initio*.

The sample molecular weight was estimated using both: (1) the SAXNS-ES server, based on the calibrated Io values (http://xray.utmb.edu/SAXNS) and (2) the saxsMoW server, which is does not depend on the accuracy of the concentration estimation, as it is uses the Porod volume to determine the molecular weight[47].

The P(r) curve or pair–distance distribution functions was determined using both GNOM from Primus[44] and the BayesApp server[48].

**Data availability.** Final coordinates of the VPS29-TBC1d5 Ins1 peptide complex are available from the RCSB with accession code 5GTU. The data sets generated during and/or analysed during the current study are available from the corresponding authors on reasonable request.

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

## Acknowledgements

We thank Drs Dominika Borek, Zhe Chen and Diana R. Tomchick for assistance with crystallographic data collection and processing. This study was supported by the Howard Hughes Medical Institute, NIH grants R01-DK073639 (E.B.), R01-AI065474 (D.D.B.) and R01-GM056322 (M.K.R), and Welch Foundation Grant I-1544 (M.K.R.). D.J. is a 'Junior One Thousand Talents' programme scholar. Results shown in this report are derived from work performed at Argonne National Laboratory, Structural Biology Center at the Advanced Photon Source. Argonne is operated by UChicago Argonne, LLC, for the U.S. Department of Energy, Office of Biological and Environmental Research under contract DE-AC02-06CH11357.

## Author contributions

D.J. conceived the project. D.J., F.L. and J.W. performed all biochemical and crystallographic work, and generated TBC1d5 KO cells. J.-S.Z., Z.D., D.G.O., C.P.-K., T.S.G., H.L. and A.S. performed cellular work. M.A.W. collected and analysed SAXS data. E.B., D.D.B. and M.K.R. supervised the research. D.J., M.A.W., D.D.B. and M.K.R. prepared the manuscript.

## Additional information

**Competing financial interests:** The authors declare no competing financial interests.

