## [Peer Review File · Nature Communications]

Reviewer #1 (Remarks to the Author):

This is a nicely conducted structure-function analysis of the retromer CSC sub-complex and TBC1d5, previously proposed to be a GAP for RAB7. The quality of the data is high and the interpretations of the data are logical and well supported. The strengths of the manuscript are (1) the clear demonstration that human TBC1d5 displays GAP activity toward Rab7-GTP, (2) an x-ray crystal structure is presented that shows how TBC1d5 binds to VPS29, and (3) the demonstration using information from the structure that TBC1d5 is essential for retromer-mediated trafficking of the MPR. The only weak point of the manuscript regards the interaction of TBC1d5 with VPS35, which is substantiated by the data, but begs more analysis. For example, do the mutations in R6 that ablate TBC1d5 binding affect RAB7 binding? Do the authors have any insight (to be added to the Discussion) regarding how the GAP activity of TBC1d5 is promoted by binding to retromer?

Reviewer #2 (Remarks to the Author):

Jia et al., provide a very comprehensive analysis of the structural basis underpinning interaction between the retromer endosomal trafficking complex and the Rab GAP TBC1d5. First they demonstrate that TBC1d5 uses its TBC GAP domain to interact with retromer with high affinity and that inserted loop sequences in TBC1d5 mediate this interaction. They next provide a detailed description of the crystal structure of TBC1d5 inserted loop 1 (Ins1) peptide in complex with the small VPS29 subunit. This work, plus other small angle X-ray scattering analyses, a low resolution crystal structure, and mutagenesis indicates that the Ins1 loop of TBC1D5 provides the primary interface to retromer through a hydrophobic surface on VPS29 and a second interface exists through the Ins2 loop and an N-terminal region of the large VPS35 subunit. This interaction is important for retromer-mediated endosomal trafficking, and appears to provide a mechanism for enhancing TBC1d5 mediated Rab7 GTP hydrolysis.

This is suggested to promote release of the retromer coat complex from the membrane allowing trafficking to occur.

Overall this paper is excellent, it is well written and begins to address very important questions in the field of endosomal trafficking, providing one of the first mechanistic insights into how regulatory proteins associate with retromer. The scope of the work is very well suited to Nature Communications and I recommend publication so long as several points can be addressed.

1. In the introduction it is discussed that the retromer core complex associates with a dimer of SNX family. It is not really clear to the reader that these are SNX family proteins with dimeric BAR

domains like SNX1, SNX2, SNX5 etc. It then goes straight on to state the retromer is recruited by SNX3, which is not a member of the SNX-BAR family proteins. This section is a bit confusing to the lay reader and needs to be rewritten for clarity to indicate the different suggested roles of SNX3 and SNX-BAR proteins.

2. For the TBC truncations used in ITC and other biochemical experiments on Page 7, can the authors comment on whether these loop-deletions affect protein folding or stability at all (e.g. were CD spectra recorded, were the protein yields similar during purification, were the monomeric by gel-filtration or aggregated?).

3. I have some concerns about the description and interpretation of the low resolution VPS35(C)-VPS29-TBC crystal structure. I found it a little confusing, and thought the main text did not align well with the structural data presented in Fig. S6. In Fig 6SA the actual structure is shown, but clearly there is no well described structural elements of TBC1d5 contacting the VPS35(C)-VPS29 complex. The figure legend states: "TBC represents the part that we could build with high confidence. Both Ins1 and Ins2 are missing in the model due to poor electron density." How poor is this density? It should be shown in Fig. 6S. It is then stated in the main text (pg. 11) that: "Within the resolution limits of the larger structure, Ins1 and VPS29 have the same conformations and make the same contacts in the Ins1-VPS29 and TBC1d5TBC-VPS29-VPS35C complexes." These two statements seem to contradict each other. If there is no interpretable density for the Ins1 loop how can you be sure that the Ins1

peptide adopts the same conformation? Then Fig 6SB shows an overlay of a TBC1d5 model generated in silico with PHYRE (which is not described in methods anywhere). In this the Ins1 loop of TBC1d5 model does not seem to look like the orientation of the actual Ins1 peptide in the VPS29-TBC1d5 Ins1 high-resolution crystal structure. I think this can all be resolved with some careful rewriting and perhaps additional figures, but certainly needs clarification.

4. The Rab GAP activity enhancement by retromer is interesting, but ideally needs some further controls. Did the authors test the retromer CSC complex on its own, both in the Rab7 GAP activity assay (i.e. does the CSC have any inherent GAP activity?) and Rab7 binding by fluorescence anisotropy (i.e. what is the affinity of the retromer CSC for Rab7 on its own, and is there cooperativity in Rab7 binding between retromer CSC and TBC1d5?).

5. In Fig. 5E and 5F the dispersal of CI-MPR localisation is used as a proxy readout for a defect in retromer-mediated CI-MPR trafficking. This data is not convincing. Previous studies indicate that retromer depletion leads to CI-MPR dispersal to punctate structures, either endosomes, or dispersed Golgi fragments. In Fig. 5E it is not possible to tell what the dispersed CI-MPR is doing. The labelling looks very diffuse, and still perinuclear. It should be compared with other markers for the TGN and endosomes (e.g. TGN38 and EEA1 or VPS35 itself).

Minor points

6. The acronym "CPC" is unnecessary. It is never used anywhere except the introduction and adds confusion with the acronym "CSC".

7. "SNX" is not defined as sorting nexin in the introduction.
8. Pg. 9 "based secondary structure" should be "based on secondary structure".
9. Pg 15. Please provide a reference for the statement regarding integrin trafficking by retromer.
10. Pg. 15. The sentences "But in cells where TBC1d5 is deleted, CD49e becomes trapped in the endosomes where retromer is located (Fig 5C). However, in cells where TBC1d5 is deleted, CD49e becomes trapped in retromer-coated endosomes (Fig 5C)," are repetitive.
11. Please provide catalogue numbers for the specific antibodies used in this study.
12. Pg. 21. Tacimate should be Tacsimate.
13. Can the panels in Fig. 5 be re-arranged as they are a little confusing? C and D should be together, and E and F should be together.

Reviewer #3 (Remarks to the Author):

This interesting study combines biochemical, structural and cell-based approaches to investigate the interaction between the core retromer complex and the TBC domain protein TBC1d5 as well as the consequences of disrupting it or depleting TBC1d5. Although a direct interaction was suspected from earlier work, this is the first characterization using purified proteins. The binding affinity and stoichiometry analyzed by ITC, and also gel filtration experiments, support the unexpected but convincing conclusion that TBC1d5 is a stable subunit of the retromer complex. Structural analyses by X-ray crystallography and SAXS provide insight into the overall architecture and high resolution details of the main binding interface between the VPS29 subunit and a peptide corresponding to an insert region in the TBC domain. The structural observations are tested by mutational analyses of residues in the main binding interface as well as the putative interface between VPS35 and a second insert

region in the TBC domain. Finally, a CRISPR knockout in HeLa cells provides evidence for a functional requirement for TBC1d5 in retromer dependent cargo trafficking. These and related findings make for a meaty contribution to understanding the structural basis for the interaction of TBC1d5 with retromer, and also provide insights into the functional properties of the TBC1d5-retromer complex. In my view, this is an important advance since little is known about the structural organization of TBC domain proteins with respect to trafficking complexes in general. The manuscript is generally well written, and I have only a few mostly minor comments.

P. 3 - "triggers hydrolysis of GTP by ARF1". Probably "... on ARF1" was meant since the GAP also contributes.

P. 6, last paragraph - it would be better to also compare Kd values if possible rather than only elution profiles on gel filtration. Differences in the latter could be due to differences in off-rates even if the affinities are similar.

P. 14 - "It is surprising that there is only a modest difference between the activity of TBC and TBC/CSC, but an apparently large difference in affinity. This could result from the nucleotide difference: GTP in the activity assay and GMPPNP in binding assay. Thus, CSC also promotes the interaction between TBC1d5 and Rab7". This explanation is confusing. I'm not aware of well documented cases of affinity differences for Rabs loaded with GTP vs. GppNHp. On the other hand, since the catalytic site in the TBC domain is distinct from the Rab7 binding site in the CSC, it would be possible for TBD1d5 binding to cause (for example) a conformational change in the CSC that substantially increases the affinity for Rab7 while having only a small perturbation of GAP activity.

P. 15 - "But in cells where TBC1d5 is deleted, CD49e becomes trapped in the endosomes where retromer is located (Fig 5C). However, in cells where TBC1d5 is deleted, CD49e becomes trapped in retromer-coated endosomes (Fig 5C), ...". Sentence repeated.

Fig. 3 - The SAXS analysis is limited to shape envelopes generated by DAMMIF without comparison with atomic resolution structures. I realize there may not be enough atomic resolution structural information for rigid body modeling with ATSAS programs such as CORAL but perhaps it might be possible to manually dock the low resolution crystal structure in Fig. 6A with the filtered DAMMIF envelopes in a plausible orientation to make the point that the two structural observations are not incompatible (e.g. is the blob that appears in the ternary complex approximately the size of the TBC domain, etc).

Fig. 5D - why is the quantification for only one TBD1d5 clone shown?

Reviewer #1:

This is a nicely conducted structure-function analysis of the retromer CSC sub-complex and TBC1d5, previously proposed to be a GAP for RAB7. The quality of the data is high and the interpretations of the data are logical and well supported.

We thank the referee for this favorable assessment of our work and manuscript.

1. The only weak point of the manuscript regards the interaction of TBC1d5 with VPS35, which is substantiated by the data, but begs more analysis. For example, do the mutations in R6 that ablate TBC1d5 binding affect RAB7 binding?

To address this question we performed GST pull-down assays between Rab7-GMPPNP or Rab7-GDP with TBC, CSC, or TBC/CSC. We found that Rab7 preferentially interacted with the TBC/CSC complex, in a GMPPNP-dependent manner. Mutation of key residues in R6 of VPS35 (VF/RR) that weakened its interaction with TBC1d5 did not alter the interaction between VPS35 with Rab7.

These data are show in new Figure S8B.

2. Do the authors have any insight (to be added to the Discussion) regarding how the GAP activity of TBC1d5 is promoted by binding to retromer?

We have added a brief speculation about this point in discussion (p. 17). It is likely that binding to CSC causes a conformational change in the TBC1d5 that increases both its affinity for Rab7 and its GAP activity. Alternatively, CSC may harbor a site that weakly binds to Rab7.

Reviewer #2:

Overall this paper is excellent, it is well written and begins to address very important questions in the field of endosomal trafficking, providing one of the first mechanistic insights into how regulatory proteins associate with retromer. The scope of the work is very well suited to Nature Communications and I recommend publication so long as several points can be addressed.

We thank the referee for this favorable assessment of our work and manuscript.

1. In the introduction it is discussed that the retromer core complex associates with a dimer of SNX family. It is not really clear to the reader that these are SNX family proteins with dimeric BAR domains like SNX1, SNX2, SNX5 etc. It then

goes straight on to state the retromer is recruited by SNX3, which is not a member of the SNX-BAR family proteins. This section is a bit confusing to the lay reader and needs to be rewritten for clarity to indicate the different suggested roles of SNX3 and SNX-BAR proteins.

We have rewritten this paragraph (p. 4) to better delineate the BAR- and non-BAR SNX proteins.

2. For the TBC truncations used in ITC and other biochemical experiments on Page 7, can the authors comment on whether these loop-deletions affect protein folding or stability at all (e.g. were CD spectra recorded, were the protein yields similar during purification, were the monomeric by gel-filtration or aggregated?).

Response: All mutant proteins have similar yield as well as chromatographic behaviors as wild type TBC1d5. All of them are monomeric on gel filtration. We have added this comment in the Experimental Procedures section on p. 20.

3. I have some concerns about the description and interpretation of the low resolution VPS35(C)-VPS29-TBC crystal structure. I found it a little confusing, and thought the main text did not align well with the structural data presented in Fig. S6. In Fig 6SA the actual structure is shown, but clearly there is no well described structural elements of TBC1d5 contacting the VPS35(C)-VPS29 complex. The figure legend states: "TBC represents the part that we could build with high confidence. Both Ins1 and Ins2 are missing in the model due to poor electron density." How poor is this density? It should be shown in Fig. 6S. It is then stated in the main text (pg. 11) that: "Within the resolution limits of the larger structure, Ins1 and VPS29 have the same conformations and make the same contacts in the Ins1-VPS29 and TBC1d5TBC-VPS29-VPS35C complexes." These two statements seem to contradict each other. If there is no interpretable density for the Ins1 loop how can you be sure that the Ins1 peptide adopts the same conformation? Then Fig 6SB shows an overlay of a TBC1d5 model generated in silico with PHYRE (which is not described in methods anywhere). In this the Ins1 loop of TBC1d5 model does not seem to look like the orientation of the actual Ins1 peptide in the VPS29-TBC1d5 Ins1 high-resolution crystal structure. I think this can all be resolved with some careful rewriting and perhaps additional figures, but certainly needs clarification.

We thank the referee for these comments. To address them we re-made/added several panels in Fig. S6. First, in a new Fig. S6B, ins1 is shown by superimposing our high resolution structure with the low resolution one. Since we do not have any structural information of Ins2, we still use a model generated in silico with PHYRE. Second, we show electron density for our overall model (Fig. S6C), around Ins1 (Fig. S6D) and Ins2 (Fig. S6E). As evident in Figure S6D, there is weak density observed for Ins1 (we apologize for the misstatement in the

previous text), and this shows good agreement with the Ins1 conformation in the high resolution structure, within the resolution limits, thus confirming our original statement that the two structures have similar orientation (Fig. S6D).

We have rephrased the text in this section to read:

TBC1d5 interacts with VPS29 only through a region corresponding to Ins1 (Fig. S6B-E). Although the electron density in this region is weak, within the resolution limits of the larger structure, Ins1 and VPS29 have the same conformations and make the same contacts in the Ins1-VPS29 and TBC1d5^{TBC}-VPS29-VPS35^C complexes.

4. The Rab GAP activity enhancement by retromer is interesting, but ideally needs some further controls. Did the authors test the retromer CSC complex on its own, both in the Rab7 GAP activity assay (i.e. does the CSC have any inherent GAP activity?) and Rab7 binding by fluorescence anisotropy (i.e. what is the affinity of the retromer CSC for Rab7 on its own, and is there cooperativity in Rab7 binding between retromer CSC and TBC1d5?).

To address this comment we performed additional Rab7 GAP activity assays (Figure 4D) and Rab7 binding assays (new Figure S8A). We did not detect any GAP activity of the CSC, even at a concentration of 320 nM. Similarly, fluorescence anisotropy assays suggests that the CSC does not interact with or binds to Rab7-GMPPNP extremely weakly.

5. In Fig. 5E and 5F the dispersal of CI-MPR localisation is used as a proxy readout for a defect in retromer-mediated CI-MPR trafficking. This data is not convincing. Previous studies indicate that retromer depletion leads to CI-MPR dispersal to punctate structures, either endosomes, or dispersed Golgi fragments. In Fig. 5E it is not possible to tell what the dispersed CI-MPR is doing. The labelling looks very diffuse, and still perinuclear. It should be compared with other markers for the TGN and endosomes (e.g. TGN38 and EEA1 or VPS35 itself).

To address these issues we have stained HeLa and the two TBC1d5 KO clones for TGN46, CD222 and VPS35 (new Figure S9). There appears to be more CD222 punctate structures, which are dispersed from the TGN46 marker in the two TBC1d5 KO clones compared to HeLa. The diffuse CD222 puncta are not localized with TGN46 (so it is also not dispersed puncta of the TGN). However, if we just look at the diffuse puncta of CD222 (which there are more in the KO cells) they are more frequently near/adjacent to VPS35.

Minor points

1. The acronym "CPC" is unnecessary. It is never used anywhere except the

introduction and adds confusion with the acronym "CSC".

We omitted the acronym "CPC".

2. *"SNX" is not defined as sorting nexin in the introduction.*

Sorting nexin is now defined at its first mention on page 4.

3. *Pg. 9 "based secondary structure" should be "based on secondary structure"*

We fixed this text.

4. *Pg 15. Please provide a reference for the statement regarding integrin trafficking by retromer.*

We have added Zech et al; Sterinberg et al; . reference supporting this statement.

5. *Pg. 15. The sentences "But in cells where TBC1d5 is deleted, CD49e becomes trapped in the endosomes where retromer is located (Fig 5C). However, in cells where TBC1d5 is deleted, CD49e becomes trapped in retromer-coated endosomes (Fig 5C)," are repetitive.*

The repetitive sentence has been deleted.

6. *Please provide catalogue numbers for the specific antibodies used in this study.*

Antibodies, including manufacturers and catalog numbers, now are listed in Supplemental Table 4.

7. *Pg. 21. Tacimate should be Tacsimate.*

We have corrected this name.

8. *Can the panels in Fig. 5 be re-arranged as they are a little confusing? C and D should be together, and E and F should be together.*

We have rearranged the panels as the referee suggests.

Reviewer #3 (Remarks to the Author):

These and related findings make for a meaty contribution to understanding the structural basis for the interaction of TBC1d5 with retromer, and also provide insights into the functional properties of the TBC1d5-retromer complex. In my view, this is an important advance since little is known about the structural organization of TBC domain proteins with respect to trafficking complexes in general. The manuscript is generally well written, and I have only a few mostly minor comments.

We thank the referee for this favorable assessment of our work and manuscript.

1. P. 3 - *"triggers hydrolysis of GTP by ARF1". Probably "... on ARF1" was meant since the GAP also contributes.*

We have changed this text according to the referee's suggestion.

2. P. 6, last paragraph - *it would be better to also compare Kd values if possible rather than only elution profiles on gel filtration. Differences in the latter could be due to differences in off-rates even if the affinities are similar.*

We agree that ideally one would compare Kd values. However, the comparison between CSC binding to TBC1d5, Fam21 and SNX3 is a very minor point in our work. Developing quantitative binding assays for these additional proteins would require significant effort, which we do not believe is warranted by the low importance of this point. Thus, we have not performed these experiments. We have, however, altered the text describing this result to indicate that the lack of coelution merely "suggests" low affinity (p. 6).

P. 14 - "It is surprising that there is only a modest difference between the activity of TBC and TBC/CSC, but an apparently large difference in affinity. This could result from the nucleotide difference: GTP in the activity assay and GMPPNP in binding assay. Thus, CSC also promotes the interaction between TBC1d5 and Rab7". This explanation is confusing. I'm not aware of well documented cases of affinity differences for Rabs loaded with GTP vs. GppNHp. On the other hand, since the catalytic site in the TBC domain is distinct from the Rab7 binding site in the CSC, it would be possible for TBD1d5 binding to cause (for example) a conformational change in the CSC that substantially increases the affinity for Rab7 while having only a small perturbation of GAP activity.

We thank the referee for this suggestion. In fact, our data do not rule out the possibility that the anisotropy changes reflect binding of Rab7 to a site on the

CSC that is altered by TBC binding, as proposed by the referee. Nevertheless, it remains possible that the differences are due more trivially to the difference between GTP and GppNHp (as anecdotally observed in our lab for other GTPase-effector interactions, which have weaker affinity for GppCp than GppNHp or GTP γ S). Because of this we have changed the text on p. 14 describing these data to raise both possibilities:

It is surprising that there is only a modest difference between the activity of TBC and TBC/CSC, but an apparently large difference in affinity. This could result from the nucleotide difference: GTP in the activity assay and GMPPNP in binding assay. Alternatively, it is possible that the changes in anisotropy reflect binding of Rab7 to a site on Vps35 in the CSC, whose affinity is enhanced by the presence of TBC1d5. Regardless of mechanism, the data indicate that the TBC1d5^{TBC}/CSC complex binds Rab7 more tightly than either TBC1d5^{TBC} or CSC alone.

P. 15 - "But in cells where TBC1d5 is deleted, CD49e becomes trapped in the endosomes where retromer is located (Fig 5C). However, in cells where TBC1d5 is deleted, CD49e becomes trapped in retromer-coated endosomes (Fig 5C), ...". Sentence repeated.

We have deleted the repetitive sentence.

Fig. 3 - The SAXS analysis is limited to shape envelopes generated by DAMMIF without comparison with atomic resolution structures. I realize there may not be enough atomic resolution structural information for rigid body modeling with ATSAS programs such as CORAL but perhaps it might be possible to manually dock the low resolution crystal structure in Fig. 6A with the filtered DAMMIF envelopes in a plausible orientation to make the point that the two structural observations are not incompatible (e.g. is the blob that appears in the ternary complex approximately the size of the TBC domain, etc).

We were able to make nice CORAL models (see below). However, we realize that the reliability of the models relies on homology modeling. In order to not over interpret the data, we did not include them in the manuscript.

Fig. 5D - why is the quantification for only one *TBD1d5* clone shown?

We have quantified data for the second *TBD1d5* clone, and they are similar to those for the first clone. These data are shown in new Figure 5D.

Reviewer #1 (Remarks to the Author):

The authors have responded favorably to all concerns raised.

Reviewer #2 (Remarks to the Author):

After reading the authors comments and revised manuscript I am happy that my main queries have been addressed and recommend publication.

Reviewer #3 (Remarks to the Author):

The authors have addressed my concerns. The manuscript is appropriate for publication without further revision.

Response to referees

As evident below, the referees asked for no additional work on the manuscript.

Reviewer #1 (Remarks to the Author):

The authors have responded favorably to all concerns raised.

Reviewer #2 (Remarks to the Author):

After reading the authors comments and revised manuscript I am happy that my main queries have been addressed and recommend publication.

Reviewer #3 (Remarks to the Author):

The authors have addressed my concerns. The manuscript is appropriate for publication without further revision.